# *k*-mer-Based Genome-Wide Association Studies in Plants: Advances, Challenges, and Perspectives

**DOI:** 10.3390/genes14071439

**Published:** 2023-07-13

**Authors:** Benjamin Karikari, Marc-André Lemay, François Belzile

**Affiliations:** 1Département de Phytologie, Université Laval, Quebec City, QC G1V 0A6, Canada; benkarikari1@gmail.com (B.K.); marc-andre.lemay.2@ulaval.ca (M.-A.L.); 2Institut de Biologie Intégrative et des Systèmes (IBIS), Université Laval, Quebec City, QC G1V 0A6, Canada; 3Department of Agricultural Biotechnology, Faculty of Agriculture, Food and Consumer Sciences, University for Development Studies, Tamale P.O. Box TL 1882, Ghana

**Keywords:** candidate genes, molecular signature, single-nucleotide polymorphism, structural variation

## Abstract

Genome-wide association studies (GWAS) have allowed the discovery of marker–trait associations in crops over recent decades. However, their power is hampered by a number of limitations, with the key one among them being an overreliance on single-nucleotide polymorphisms (SNPs) as molecular markers. Indeed, SNPs represent only one type of genetic variation and are usually derived from alignment to a single genome assembly that may be poorly representative of the population under study. To overcome this, *k*-mer-based GWAS approaches have recently been developed. *k*-mer-based GWAS provide a universal way to assess variation due to SNPs, insertions/deletions, and structural variations without having to specifically detect and genotype these variants. In addition, *k*-mer-based analyses can be used in species that lack a reference genome. However, the use of *k*-mers for GWAS presents challenges such as data size and complexity, lack of standard tools, and potential detection of false associations. Nevertheless, efforts are being made to overcome these challenges and a general analysis workflow has started to emerge. We identify the priorities for *k*-mer-based GWAS in years to come, notably in the development of user-friendly programs for their analysis and approaches for linking significant *k*-mers to sequence variation.

## 1. Use of Genome-Wide Association Studies in Crops

Genome-wide association studies (GWAS) have routinely been used in plant science for the discovery of significant markers, candidate genes, and beneficial alleles, as well as understanding the genetic architecture of traits of economic importance [1,2,3]. Put simply, GWAS analyses involve determining the statistical relationship that may exist between phenotypes (traits) and molecular markers. Such analyses have become mainstream over the last decade as a result of the availability of crop reference genomes, drastic reductions in genotyping costs, the development of more powerful statistical models, and advancements in bioinformatics tools [4,5,6,7]. In addition, the advent and application of new phenotyping platforms have increased the throughput and improved data accuracy compared to conventional phenotyping [8].

The GWAS provides several advantages over its sister mapping strategy, linkage mapping. Notably, the GWAS requires less time to assemble the mapping population, provides higher resolution and more recombination events, facilitates the identification of candidate genes, and generates results that can be transferred more easily to marker-assisted breeding [4,9]. For example, Li, et al. [10] developed a derived cleaved amplified polymorphic sequence (dCAPS) marker for a *SW9-1* (ss246792949T/C) locus associated with soybean seed size, and the dCAPS marker was able to discriminate between accessions with small and large seeds.

The most commonly used markers in GWAS are single-nucleotide polymorphisms (SNPs), which can be easily discovered and genotyped on the genome scale using various array- or sequencing-based genotyping platforms [11]. SNP arrays are affordable and convenient, as they can be designed once and manufactured in as many copies as needed [12]. However, they typically rely on a single reference genome and a few additional accessions, which constrains the set of SNPs that can be genotyped from them [13]. Sequencing-based approaches are more flexible and can provide a genome-wide assessment of variation if whole-genome sequencing (WGS) data are available [14]. However, many sequencing-based GWAS analyses rely on methods such as genotyping-by-sequencing (GBS) to genotype fewer markers on a larger number of samples (e.g., [15]).

## 2. Limitations of Current GWAS Methods

Although the GWAS has proved useful for quantitative trait loci (QTL) discovery in crop species, the task of identifying candidate genes and causal variants from the output of GWAS is still difficult [16,17]. As a result, GWAS analyses often fail to pinpoint the causal genes or variants responsible for variation in a particular trait. Although this failure to pinpoint causal variation could be addressed through various methodological improvements, the use of SNPs as a single type of variant in GWAS is an obvious limitation. Indeed, SNP datasets are typically based on mapping to a single reference genome, which may lead to inaccurate genotyping or the exclusion of genomic regions that are not represented in the reference genome, particularly in complex genomes like maize, soybean, and wheat [18,19,20,21,22].

Some limitations of SNP-based GWAS could be overcome using different types of genotypic data [23]. For example, many studies have revealed that large structural variations (SVs) are more likely to have phenotypic impacts than SNPs [24,25,26,27,28]. In contrast to SNPs, SVs are large variations involving a difference of at least 50 nucleotides between a reference sequence and an alternative sequence [29]. These variants can take various forms, such as insertions, deletions, duplications, copy number variants (CNVs), or translocations [30,31]. The pioneering report of genic SVs affecting a phenotype dates back almost a century, when it was discovered and documented that a duplication of the *Bar* gene is linked to small eyes in fruit flies [32]. Over the last decade, it has been demonstrated that SVs affect traits in plants such as shoot architecture, flowering time, fruit size, and stress tolerance [30,33,34,35,36,37,38,39]. SVs are reported to influence gene expression in various ways, such as disruption to gene structure, alteration of CNV, or the composition/positioning of *cis*-regulatory sequences [40]. The potential of a greater functional impact has motivated the use of SVs instead of SNPs for GWAS in crops such as cotton [26], soybean [41], cucumber [42], and tomato [43], an approach which we refer to here as an SV-based GWAS.

Despite the relevance of SVs, the identification of SVs from short-read sequencing is difficult and unreliable, leaving the majority of SVs poorly resolved and their molecular as well as phenotypic impacts largely overlooked [31,44,45]. As a consequence, population-scale assessments of variation in plants are disproportionally skewed toward SNPs and small insertions and/or deletions (indels) [30,46]. Despite increasing population-scale reports of SV in several crop species in the last few years [37,47,48,49,50,51], it is still difficult to obtain accurate SV genotype data on the scale required for GWAS.

In summary, SNP datasets are not sufficient to describe the whole spectrum of sequence variation and SVs are difficult to accurately discover and genotype on the population scale. Therefore, methods that enable researchers to encompass all types of variation without the pitfalls of genome-wide SV discovery are much needed. To overcome these limitations, *k*-mer-based GWAS approaches that are agnostic to variant types and reference-free have been developed [52,53,54]. The subsequent sections in this review will focus on *k*-mer-based GWAS approaches, examples of use in crops, and specific challenges, as well as future developments and perspectives.

## 3. The Concept of *k*-mer-Based GWAS

*k*-mers are subsequences of a fixed length *k* which can be obtained from sequencing data or genome assemblies by extracting all such subsequences found in the input dataset. For example, from the sequence ACCGTCG, the following *k*-mers of length four (4-mers) can be observed: ACCG, CCGT, CGTC, and GTCG. *k*-mers have been used for various applications in genomics, notably genome assembly [55], alignment-free sequence comparison [56], and variant genotyping [57,58]. In the context of GWAS, *k*-mers can be used to identify statistical associations between the occurrence of *k*-mers in a dataset and traits of interest. In its simplest implementation, the *k*-mer-based GWAS relies on a presence/absence table of *k*-mers found across various accessions (e.g., [54,59]). In this case, the output of the analysis indicates whether the presence of a given *k*-mer is associated with phenotypic variation. In more complex models, the number of times that a given *k*-mer is observed is used for modelling the statistical relationship between *k*-mers and phenotypic observations (e.g., [52]). In all cases, the output of the *k*-mer-based GWAS is a *p*-value indicating the probability that a given *k*-mer is associated with the trait being studied.

*k*-mer-based GWAS analyses work because variation in genomic sequences between individuals ultimately results in differences in the occurrence of *k*-mers in sequencing data (Figure 1). For example, if two individuals harbor different genotypes in an SNP locus, *k*-mers overlapping with the SNP will differ. These differences in *k*-mer occurrence patterns may theoretically occur due to any kind of sequence variation [53]. In practice, given that *k*-mers are typically quite short (25- to 51-mers are typical for use in GWAS), variation in repetitive regions may not result in *k*-mer patterns that can be observed in typical short-read WGS data. Also, although *k*-mer-based GWAS could theoretically be applied to reduced-representation sequencing methods such as GBS, their full potential to pinpoint causal variants will only be reaped using WGS data.

*k*-mer-based GWAS provide several advantages over conventional SNP-based GWAS. First, *k*-mers are agnostic to the nature of the sequence variation underlying differences in their occurrence [24,60,61]. This provides a universal way to assess variation due to SNPs, indels, and SVs in GWAS without having to specifically discover and genotype these variants. Second, *k*-mer-based analyses do not depend on a reference genome and can therefore be used in species that lack a reference genome (e.g., [22]). Even for species that do have a reference genome, *k*-mer-based analyses allow the detection of significant *k*-mers originating from regions that are absent from the reference, such as large insertions (e.g., [54]). Third, using *k*-mers derived directly from raw sequencing data avoids potentially error-prone variant discovery and genotyping steps. This facilitates the identification of causal variants with *k*-mers as compared to SNP- or SV-based GWAS (e.g., [62]). Finally, variants that occur at a distance of at most *k* nucleotides can be captured within the same *k*-mers and therefore provide a way to assess phenotypic differences due to haplotypes rather than single markers (see, e.g., [54]).

## 4. Methods Used in *k*-mer-Based GWAS

The main disadvantage of *k*-mer-based GWAS approaches is the lack of standardized methods for conducting such analyses. Indeed, these methods are relatively recent and therefore have not yet matured to the point where state-of-the-art analysis pipelines have emerged. As a result, user-friendly programs such as TASSEL [63], mrMLM.GUI [7], and GAPIT [64] have been developed for conventional GWAS analyses using SNPs. On the contrary, much work remains to be conducted in making *k*-mer-based GWAS programs user-friendly for wider application in the plant genomics community. Nevertheless, a general workflow for conducting *k*-mer-based GWAS has started to emerge (Figure 2). In this section, we will review this general workflow and discuss how it has been applied in previous studies.

The first step in conducting *k*-mer-based GWAS is to count the *k*-mers occurring in sequencing data (Figure 2a). This step is computationally straightforward, as several efficient bioinformatics tools have already been developed for this task (see [65,66,67,68,69]). Still, care must be taken in choosing parameters for counting and filtering *k*-mers [61]. Studies published to date have used various *k*-mer lengths (25-mers to 51-mers) for GWAS, but the thorough testing of a wide range of *k*-mer lengths has yet to be performed. Shorter *k*-mers reduce the probability of including sequencing errors and result in more *k*-mers being tallied for a given dataset, therefore providing higher counts for resolving copy number variants based on *k*-mer counts. On the other hand, longer *k*-mers would provide a larger number of unique *k*-mers and more power to resolve sequences stemming from repetitive regions. In addition to properly choosing the *k*-mer length, raw *k*-mer counts must be filtered prior to downstream analyses. Typically, *k*-mers that are only observed once in a sample are discarded as they might be due to sequencing errors (Figure 3a). Moreover, Voichek and Weigel [54] advocate for *k*-mers to be filtered based on their occurrence on both strands. Indeed, a *k*-mer that is only ever observed on one strand (i.e., it is not observed as its reverse complement) may indicate contamination, e.g., due to sequencing adapters. *k*-mers may be further filtered based on minor allele frequency, as in standard GWAS methods.

Once *k*-mers are counted and filtered, the *k*-mer-based GWAS analysis per se can be computed (Figure 2b). The most straightforward approach is to conduct GWAS based on the presence/absence of *k*-mers, as in Voichek and Weigel [54] or Gaurav et al. [59]. In such analyses, the presence of a *k*-mer is considered an allele, whereas its absence is considered another allele; standard GWAS approaches used for SNPs can therefore be applied to such datasets. Voichek and Weigel [54] used the GEMMA software [70] to compute a mixed linear model (MLM) using a kinship matrix directly computed from the *k*-mer presence/absence table. Given the large number of *k*-mers involved (can be tens of millions), their approach first computes an approximate score for each *k*-mer in the dataset. The most significant *k*-mers identified using the approximate scores are then used to compute the exact *p*-values in GEMMA. Gaurav et al. [59] used a different approach for reducing the number of analyzed *k*-mers by keeping only those whose presence/absence pattern was correlated to phenotypic data above some preset threshold; exact *p*-values for the filtered *k*-mers were then computed using linear regression models, accounting for population structure. Presence/absence-based approaches are convenient because the minimum average sequencing depth required to attain low error rates in determining presence/absence is rather low. Using empirical data [62], and assuming that sequencing depth follows a Poisson distribution, we estimate that a sequencing depth of 10 to 15X will be sufficient for the purposes of most *k*-mer-based GWAS relying on presence/absence (Figure 3b,c).

**Figure 2 genes-14-01439-f002:**
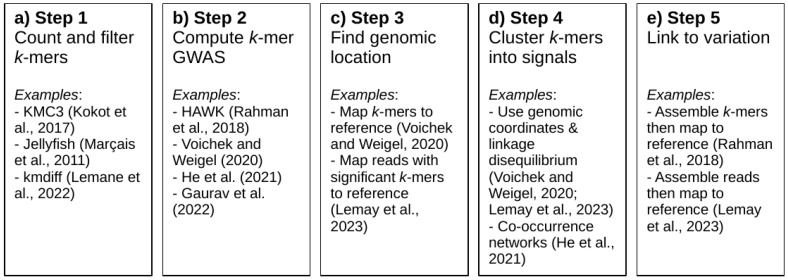
Summary of the analytical steps involved in *k*-mer-based GWAS with examples of how these steps were carried out in studies published to date. (**a**) Examples of bioinformatics software that can be used to count *k*-mers in sequencing data. (**b**) List of studies that have developed software for computing *k*-mer-based GWAS. (**c**) Finding the genomic location of significant *k*-mers is typically carried out by mapping *k*-mers or reads containing them to a reference genome (when available). (**d**) Clustering *k*-mers into signals has been conducted in various ways and can help in identifying the number of loci controlling a trait or filtering out spurious associations. (**e**) Linking significant *k*-mers to sequence variation is typically achieved through the assembly of significant *k*-mers or reads containing them, but much work remains to be performed in developing systematic approaches for doing so. [52,53,54,59,62,65,67,71].

**Figure 3 genes-14-01439-f003:**
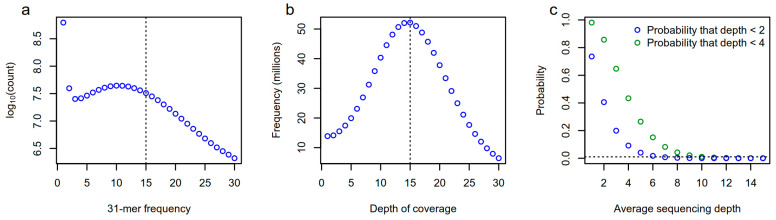
Implications of sample sequencing depth for *k*-mer-based GWAS. (**a**) Frequency distribution of *k*-mers of length 31 (31-mers) in a soybean sample analyzed by Lemay et al. [62], showing a clear peak for k-mers observed only once or twice due to sequencing errors. Note the logarithmic scale on the *y*-axis. The vertical dotted line indicates the average mapping depth of the sample shown (15X). (**b**) Distribution of the number of times a given genomic position is covered by sequencing data in that same sample. In the sample shown, 1.5% of positions are observed only once. The vertical line again indicates the average mapping depth. (**c**) Theoretical probability that a given position is observed less than two (blue circles) or four (green circles) times for a given average sequencing depth by assuming a Poisson distribution with lambda equal to the average sequencing depth. According to these distributions, sequencing depths of 7 and 11 ensure a probability < 1% (horizontal dotted line) to falsely declare a given position as “absent” based on a minimum of 2 or 4 reads, respectively. While sequencing data do not exactly obey a Poisson distribution, it is safe to assume that a minimum sequencing depth of 10 to 15X will be sufficient for most GWAS applications relying on the presence/absence of a *k*-mer.

More elaborate *k*-mer-based GWAS approaches use *k*-mer counts instead of their mere presence/absence patterns for GWAS [52,53]. These models are potentially more powerful than the method developed by Voichek and Weigel [54], as they can distinguish between heterozygous genotypes and variants (such as copy number variants) that would not be detected through presence/absence alone. Rahman et al. [53] proposed a method that used *k*-mer counts but could only analyze qualitative traits based on the comparison of *k*-mer counts in case and control groups. On the other hand, He et al. [52] proposed the first method based on *k*-mer counts for analyzing quantitative traits using linear regression models controlling for population structure. They also tested various transformations of the *k*-mer counts and compared their impacts on the results. Despite their great potential, there are pitfalls associated with count-based methods, as *k*-mer counts will vary stochastically due to sequencing technology bias (e.g., the GC content of reads may affect sequencing depth). Count-based methods will therefore need to rely on deeper sequencing to accurately resolve the statistical relationship between *k*-mer copy number and phenotypic variation.

*k*-mer-based GWAS analyses truly differ from conventional GWAS when the time comes to analyze significant signals (Figure 2c). While the genomic position of molecular markers used in conventional GWAS is typically known, the output of *k*-mer-based GWAS consists solely of a list of *k*-mers and their associated *p*-values. Consequently, the burden of associating these *k*-mers with genomic positions must be handled in downstream analyses, for which no standard method exists yet. The most straightforward approach for linking significant *k*-mers to their genomic location is to map them to a reference genome, and it is indeed the approach used by most studies published so far (e.g., [53,54]). Once the genomic positions are known, results can be displayed using Manhattan plots as in standard GWAS analyses, except that results are typically only displayed for significant *k*-mers instead of genome-wide ones. A slightly different approach was used by Lemay et al. [62], who developed the katcher program (https://github.com/malemay/katcher) to extract all of the sequence reads containing significant *k*-mers. This approach is more exhaustive than simply mapping *k*-mers to a reference because it considers the entire read sequence for mapping purposes.

Another problem that is somewhat related to finding the position of significant *k*-mers lies in clustering them into coherent signals (Figure 2d). Indeed, the immediate output of *k*-mer-based GWAS analyses tells us nothing about the number of significant loci identified. An obvious approach, which is analogous to what is the case in conventional GWAS, consists of grouping *k*-mers that can be mapped in contiguous regions of a reference genome. This approach is implied by the peaks observed in Manhattan plots and was used by Voichek and Weigel [54] and by Lemay et al. [62] to identify the number of loci found by a given GWAS analysis. However, this approach alone is hardly sufficient because *k*-mers may not map onto the reference genome or may appear to be mapped in different regions and yet belong to the same locus. Published *k*-mer-based GWAS analyses have therefore also typically used some form of clustering to group co-segregating significant *k*-mers together. For example, Voichek and Weigel [54] and Lemay et al. [62] computed linkage disequilibrium between *k*-mers to identify the number of loci found by a given analysis. This method has also allowed Lemay et al. [62] to determine that apparently spurious significant hits found for some traits actually co-segregated with genuine loci. He et al. [52] instead used co-occurrence networks to identify co-segregating *k*-mers.

Identifying the variation underlying significant k-mers is likely the most difficult aspect of conducting *k*-mer-based GWAS (Figure 2e). Indeed, the fact that *k*-mers can act as a molecular signature for all kinds of sequence variation is a double-edged sword: while it makes the approach universal, it also makes it difficult to link significant *k*-mers to biologically meaningful variation. This problem has typically been tackled through the assembly of significant *k*-mers [53] or reads containing them [54,62]. These assemblies are then compared to a reference assembly to identify the sequence variation implied by significant *k*-mers (Figure 4a,b). However, this approach is time-consuming and difficult to generalize. Efficient, standardized methods for identifying sequence variation from *k*-mers are therefore much needed.

It is important to note that only steps 1 and 2 (counting *k*-mers and computing GWAS) of our summarized framework are truly mandatory; steps 3 to 5 (finding genomic location, clustering *k*-mers into signals, and linking *k*-mers to sequence variation) are optional to some extent. For example, in species that lack a reference genome, step 3 is not possible, but significant *k*-mers can nevertheless be clustered and/or assembled to gather meaningful information. For example, in a study of resistance to wheat stem rust in *Aegilops tauschii*, for which no reference genome was available, Arora et al. [22] focused on nucleotide-binding/leucine-rich repeat (NLR) genes through the use of sequencing data enriched in such sequences. Significant *k*-mers were subsequently mapped to local assemblies of NLR genes generated from the sequencing data themselves. For such analyses involving species without a reference genome, the local assembly of significant *k*-mers or reads containing them will likely be necessary to obtain meaningful results. Step 5 (linking *k*-mers to sequence variation) may however not be necessary at all to obtain meaningful results from *k*-mer-based GWAS. For example, in a set of studies involving *Ae. tauschii* [22,59] and wheat [73], the functional impact candidate genes could be demonstrated without the causal variant itself being identified at the sequence level.

## 5. Case Studies of *k*-mer-Based GWASs in Plants

Applications of *k*-mer-based GWAS are limited in plants compared with prokaryotic organisms [53,74,75,76,77]. This section highlights empirical results obtained via *k*-mer-based GWAS conducted in plants, with an emphasis on their findings and key features (Table 1).

In the first study reporting *k*-mer-based association in plants, Arora et al. [22] phenotyped 151 *Aegilops tauschii* accessions from diverse sources against six different races of *Puccinia granminis* f. sp. *tritici* (PGT). Given the expectation that genes associated with resistance to this pathogen would be NLR genes, they performed their analysis on sequencing data enriched for NLR genes, an approach that they named AgRenSeq. In a presence/absence-based strategy that involved pre-filtering based on a correlation with the trait and linear regression models accounting for population structure, 51-mers were used to derive *p*-values. Candidate genes were identified by mapping significant *k*-mers to locally assembled NLR gene sequences. This analysis readily identified four NLRs that were consistent with the results published in previous studies [78,79,80,81], and two of them were validated using transformation. While the study published by Arora et al. [22] was not truly genome-wide because sequence enrichment was used, it still demonstrated the use of *k*-mers for association mapping in plants.

**Table 1 genes-14-01439-t001:** k-mer-based GWAS papers in plant science.

Crop	Number of Genotypes	Trait	Length of k-mer	Key Feature	Reference
Wild diploid wheat (*Aegilops tauschii*)	195 (151 were used for phenotyping)	Stem rust (caused by *Puccinia graminis* f. sp. *tritici*)	51	Used sequencing data enriched for NLR genes instead of a genome-wide approach.	[22]
*Arabidopsis*	1135	Germination, seedling growth, flowering time, etc.	25, 31	Discovered new associations with structural variants and with regions missing from reference genomes.	[54]
Tomato	246	Days to tassel, ear weight, etc.			
Maize	282	96 metabolites, including guaiacol			
Soybean	438 *Gylcine* accessions	Seed pigmentation	31	*k*-mer-based approach mapped genomic region for recombinant event at *I* locus.	[27]
Maize	282	Upper leaf angle, flowering time, cob and kernel color, and seed oil content	25, 31	Used whole-genome sequencing data from the Maize 282 Association Panel (maize282) [82] to conduct both *k*-mer- and SNP-based GWAS.	[52]
Wild diploid wheat *(Aegilops tauschii)*	242	Stem rust (caused by *P. graminis* f. sp. *tritici*), powdery mildew (caused by *Blumeria graminis* f. sp. *tritici*), resistance to the wheat curl mite *Aceria tosichella* (vector of wheat streak mosaic virus), leaf trichomes, flowering time, and spikelet number per spike	51	Genome-wide extension of the method developed by Arora et al. [22].	[59]
Wheat	320, including 300 landraces	Blast fungus (caused by *Pyricularia oryzae*)	51	Functional validation of a candidate gene via virus-induced gene silencing and development of functional markers.	[73]
Soybean	363 *G. max*	13 traits including pod color, pubescence form, and resistance to the oomycete *Phytophthora sojae*	31	Detected several well-known loci/genes for each of the traits.	[62]

Voichek and Weigel [54] used 1135, 246, and 282 accessions of *Arabidopsis*, tomato, and maize, respectively, to conduct *k*-mer- and SNP-based GWAS with several hundreds of traits (Table 1). In their study, the associations detected via the *k*-mers largely overlapped with the results from the SNP-based approach. The former approach had stronger statistical support, with *p*-values being generally more significant for *k*-mers than for SNPs at a locus. In addition to the ability of *k*-mers to detect genomic regions found via SNP-based GWAS, they detected short indels and SVs, as well as signals in regions outside of the reference genomes of *Arabidopsis* and tomato. Nevertheless, some significant associations were only detected through SNP-based analysis, suggesting that SNP-based analysis may be complementary to *k*-mers.

Kim et al. [27] mapped the genetic basis of seed pigmentation traits in soybean with SNP- and *k*-mer-based GWAS. Their *k*-mer-based analysis was focused on the *I* locus as the SNP-based approach failed to dissect the detailed genomic structure of the *I* locus due to incomplete assembly around the corresponding region. The *k*-mer approach highlighted a recombination event at the *I* locus that explained the occurrence of yellow seed coats. Although limited in scope in this study, the *k*-mer-based approach yielded a comparative advantage over the SNP-based approach by allowing the detection of this complex variant.

He et al. [52] used the Maize 282 Association Panel (maize282) from Flint-Garcia, et al. [82] to conduct *k*-mer- and SNP-based GWAS on upper leaf angle, flowering time, cob and kernel color, and seed oil content (Table 1). A number of known loci/candidate genes were identified via *k*-mer-based GWAS, whereby most of which were detected using SNP-based GWAS. Nevertheless, an additional *Pericarp color 1* homologous gene, *Pericarp color 2*, was found via associated *k*-mers for cob color, but not detected via SNP-based GWAS. Several other known genes such as *DGAT1-2* (for kernel oil), *yellow endosperm 1, carotenoid cleavage dioxygenase 1*, and *zeaxanthin epoxidase 1* (kernel color) [52] were detected. They also provided an integrative analysis by integrating their *k*-mer analysis with gene expression data and used *k*-mers to predict phenotypes.

Gaurav et al. [59] extended the method developed by Arora et al. (2019) to the whole genome in their analysis of 242 *Ae. Tauschii* accessions. In their study, the significantly associated *k*-mers for stem rust (caused by *P. graminis* f. sp. *tritici*) were mapped to both the *Ae. tauschii* AL8/78 reference genome and a de novo assembly of a relevant accession (with two cloned stem rust resistance genes), which was anchored to the reference genome. In addition, they identified a number of candidate genes which had previously been demonstrated to influence the studied traits. These include stem rust (*SrTA1662*) [22,79], flowering time (*FLOWERING LOCUS T1*) [83,84,85], leaf trichomes (*RAMOSA3* and *TPP4*) [86], spikelet number per spike (SISTER OF *RAMOSA3*) [87], and wheat curl mite (*Cmc3* and *Cmc4*) [88,89]. Interestingly, they detected no NLR immune-receptor-encoding gene for powdery mildew resistance, but identified an insertion containing a wheat-tandem kinase (WTK) belonging to a gene family known to confer resistance to wheat stripe rust (*Yr15*) [90], stem rust (*Rpg1* and *Sr60*) [91,92], and powdery mildew [93].

Arora et al. [73] used 320 wheat lines (including 300 landraces) to screen with Br48 a strain of *Pyricularia oryzae* (alternatively called *Magnaporthe oryzae*), transformed with either *PWT3* or *PWT4* from a previous study [94]. They conducted *k*-mer-based association mapping based on NLR-enriched sequencing data, from which they identified two candidate genes, *TraesCS1D02G029900* and *TraesJAG1D03G00423690*, for *PWT3* and *PWT4,* respectively. These genes, respectively, overlapped with *Rwt3* [95] *and Rwt4* [96] from a biparental mapping study. To validate these candidate genes, further screenings were conducted and different types of mutations were identified in *TraesCS1D02G029900* (*Rwt3*) in a TILLING population of Jagger [97]. A virus-induced gene silencing experiment was further conducted to functionally validate *TraesCS1D02G029900* (*Rwt3*), which showed that this gene is required for resistance to *P. oryzae*, expressing the *PWT3* effector. They subsequently developed kompetitive allele-specific PCR markers (for both *Rwt3* and *Rwt4*), which they tested in the landrace subpopulation, and obtained a validation rate of 97 and 99%, respectively. The functional validation results from this study provide compelling evidence for the value of *k*-mer application in GWAS mapping.

A recent study by Lemay et al. [62] from our research group conducted SV-, SNP-, and *k*-mer-based GWAS with 363 *Glycine max* accessions for 13 traits including flower, pubescence, seed coat, and hilum color, as well as stem termination type, pubescence density, and seed coat luster (Table 1). A number of well-known validated loci/genes were detected, including the *W1* locus/*Glyma.13g72100* for flower color [98], the *T*/*Glyma.06g202300* [99] and *Td*/*Glyma.03g258700* [100] loci for pubescence color, and the *G* locus/*Glyma.01g198500* for seed coat color [72], among others. Of the three GWAS approaches, the *k*-mer-based strategy proved to be the best at pinpointing causal variants based on the most significant *k*-mers. Moreover, using the combined results of the SNP- and *k*-mer-based approaches, they were able to suggest novel candidate genes at some loci. This study therefore advocates for the complementarity of SNP- and *k*-mer-based approaches in identifying candidate genes.

So far, these seven *k*-mer-based GWAS papers highlight the power of this strategy compared to other known strategies (SV- and SNP-based), and warrant application in plant science to discover novel loci/candidate genes for traits of economic importance.

## 6. Challenges and Perspectives

A number of challenges have been reported in the *k*-mer-based GWAS papers published to date [24,52,54,62]. Some prominent challenges are data size and complexity, which not all research teams may be prepared to deal with. For example, He et al. [52] detected a total of 1.1 billion non-redundant *k*-mers from 261 inbred lines in their study, with the majority of lines having 0.35–0.55 billion non-redundant *k*-mers. This can lead to challenges in data storage, processing, and analysis. Therefore, large *k*-mer datasets can be computationally intensive and require significant computational resources, such as dedicated high-performance computing servers. Efficient algorithms and programs designed specifically for the analysis of *k*-mer-based GWAS have therefore been developed (e.g., [52,54,62]) and will still need to be developed in years to come.

A related problem is the development of user-friendly programs for conducting *k*-mer-based GWAS and associated analyses. To date, the programs available for *k*-mer-based GWAS have largely targeted an audience who is already proficient with programming and bioinformatics. Efforts to bring these programs to a wider audience have been made by the likes of kGWASflow [101], kmdiff [71], and gwask [62]. Still, much work remains to be conducted to make these methods available to the community. In particular, packages written in popular scripting languages (e.g., R, Python) should be developed to help with the analysis and visualization of the results of *k*-mer-based GWAS.

Another challenge faced by *k*-mer-based GWAS analyses is the appropriate control of false positive associations. While a *k*-mer-based kinship matrix was successfully used by Voichek and Weigel [54], this approach may not be sufficient to prevent the spurious associations that could arise from using *k*-mers instead of conventional molecular markers. For example, Lemay et al. [62] found spurious associations scattered throughout the genome for some of the traits that were analyzed. The fact that these spurious associations had less significant *p*-values than *bona fide* loci suggests that work remains to be conducted in appropriately choosing significance thresholds for *k*-mer-based GWAS. These spurious associations may also have been due to the presence of repetitive sequences, which resulted in some significant *k*-mer mapping to apparently random locations throughout the genome.

As discussed above, the biggest improvement that has yet to be made in *k*-mer-based GWAS lies in the identification of sequence variation underlying significant *k*-mers. Assembly-based approaches used to date have been useful for some purposes but have not provided a universal method to link significant *k*-mers to biologically meaningful variation. Going forward, we believe that approaches based on pangenome graphs [57,102,103,104] hold great promise. Such approaches have been identified as a priority for further research in plant genomics [23,40] but have yet to be applied to the analysis of *k*-mer-based GWAS. Pangenome graphs could facilitate the analysis of significant *k*-mers by allowing their alignment to the graph and the identification of haplotypes associated with phenotypic variation.

We anticipate that future developments in sequencing technology should enhance the use of *k*-mers in GWAS. So far, sufficient sequencing depth and sequence quality for *k*-mer-based analyses have only been provided by Illumina short-read sequencing technology. As a result, *k*-mer length for use in GWAS is limited to a few dozen nucleotides, whereas *k*-mer lengths in excess of a hundred nucleotides would provide a much greater resolution of variants located in repetitive regions. The relatively high error rates of current long-read technologies (PacBio and Oxford Nanopore technologies, [105]) prohibit the extraction of long *k*-mers from their sequences, as the probability of including errors in such *k*-mers would be extremely high. One exciting development is PacBio HiFi data [106], which can deliver highly accurate long reads; however, this method is still too costly to apply on the population scale required for GWAS. In the meantime, the use of long-read correction methods [107] may be an interesting avenue for the use of long-read sequencing data in *k*-mer. Overall, given the pace of recent developments in long-read technologies, the moment when long-read sequencing will fuel *k*-mer-based GWAS efforts is probably not too far away.

Beyond methodological improvements in GWAS analysis, we believe that the use of *k*-mer-based association methods in years to come will prove to be transformative for the discovery of new gene–trait associations. As discussed in this review, *k*-mer-based methods have already provided novel insights into crops such as soybean, maize, and wheat. Going forward, we believe that such approaches will greatly enhance our understanding of crop genetics and ultimately our ability to develop crop varieties that will tackle the challenges associated with population growth and climate change.

## Figures and Tables

**Figure 1 genes-14-01439-f001:**
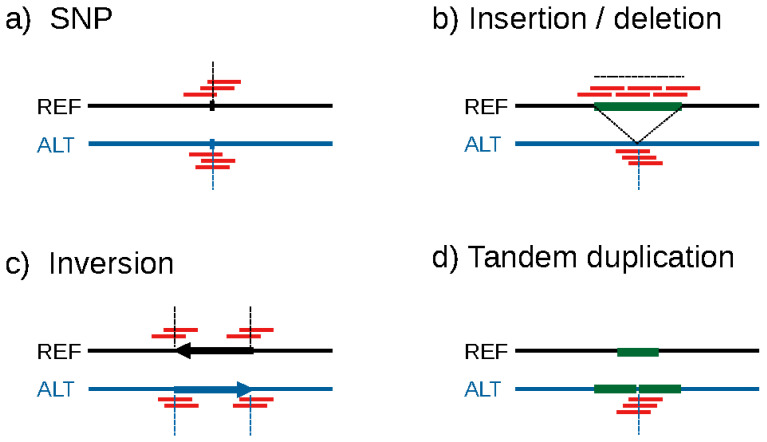
Illustration of the location of unique *k*-mers (red lines) originating from a reference genome (solid black lines) and an alternate genome (solid blue lines) depending on the underlying variant type. Dashed lines indicate genomic locations (vertical lines) or ranges of genomic positions (horizontal lines) that will induce unique *k*-mer patterns in the respective sample. (**a**) A single-nucleotide polymorphism (SNP, indicated by ticks on the genomes) will result in *k*-mers specific to each genome when *k*-mers overlap with the SNP. (**b**) For insertions/deletions, unique *k*-mers will originate from the breakpoints induced by the variation. In addition, if the sequence insertion is novel (i.e., not found elsewhere in the genome), unique *k*-mers will also originate from within the inserted sequence. (**c**) For inversions, unique *k*-mers arise at the inversion breakpoints. *k*-mers originating from within the inverted sequence will not differentiate between the two genomes because only the orientation of the sequence will differ. (**d**) For tandem duplications, novel adjacencies will only occur in the genome bearing the additional copies. However, *k*-mer counts may be able to differentiate between varying copy numbers at the locus.

**Figure 4 genes-14-01439-f004:**
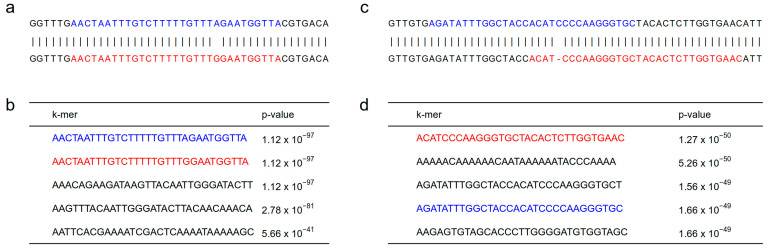
Example of identification of sequence variants underlying significant *k*-mersat soybean loci. (**a**) Pairwise alignment between two haplotypes generated from the local assembly of reads containing significant *k*-mers at the *G* locus controlling seed coat color. The causal variant at this locus is A > G SNP [72]. (**b**) Table showing the most significant *k*-mers identified by Lemay et al. [58] at this locus and their associated *p*-values, sorted from the most to least significant. The colors of the two most significant *k*-mers match their location in the haplotypes in panel (**a**). (**c**) Pairwise alignment between two haplotypes generated from local assembly of reads containing significant *k*-mers at the T locus controlling pubescence color. The causal variant at this locus is 1-bp indel. (**d**) Table showing some significant *k*-mers identified by Lemay et al. [58] at this locus and their associated *p*-values, sorted from the most to least significant. The red and blue color of two significant *k*-mers match their location in the haplotypes in panel (**c**).

## Data Availability

No data were generated in the compilation of this review.

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
