# Peer review of "k-mer-Based Genome-Wide Association Studies in Plants: Advances, Challenges, and Perspectives"

_genes, 2023, doi:10.3390/genes14071439_

Round 1
Reviewer 1 Report
Karikari, Lemay and Belzile’s have written a welcome review on the application k-mer GWAS in plants, summarizing eloquently the advantages of k-mer GWAS and the challenges that lie ahead. The application of k-mers in genomics and population genetics is a rapidly emerging field – indeed one colleague of ours recently said “you can do almost anything with k-mers these days.” The review is well written, and the figures illustrate in an elegant and simple way some of the quintessential advantages of k-mer GWAS. We therefore expect the review will find traction amongst the scientific community studying plant genetics, genomics, and population biology. Notwithstanding, we do have some concerns, as listed below, which we would advise be addressed :
Major concern:
Surprisingly, for such a new and emerging field, the review fails to make any mention of the pioneering work by Arora et al., 2019, Gaurav et al., 2022, and Arora et al., 2023. Arora et al., 2019 were the first to employ k-mer GWAS in plants, the first to demonstrate the use of k-mer GWAS without a reference genome, and the first to use the technology to clone plant genes. This feat was repeated in Gaurav et al., 2022, and in Arora et al., 2023. The three studies demonstrated the use of k-mer GWAS in two different species, the wild wheat progenitor Aegilops tauschii as well as hexaploid bread wheat, Triticum aestivum. To our knowledge, no other study has to date used k-mer GWAS to identify new genes for which the function was unequivocally confirmed using transgenics, mutants, and/or VIGS.
By not mentioning these studies in their review, Karikari, Lemay and Belzile falsely attribute ‘first’ achievements and critical advances to other studies and make false conclusions about the length of k-mers used in previous studies. We suggest the authors carefully go through their review, re-consider Table 1, and rewrite these critical statements considering this trio of studies.
Arora, S., Steuernagel, B., Gaurav, K. et al. Resistance gene cloning from a wild crop relative by sequence capture and association genetics. Nat Biotechnol 37, 139–143 (2019). https://doi.org/10.1038/s41587-018-0007-9
Gaurav, K., Arora, S., Silva, P. et al. Population genomic analysis of Aegilops tauschii identifies targets for bread wheat improvement. Nat Biotechnol 40, 422–431 (2022). https://doi.org/10.1038/s41587-021-01058-4
Arora, S., Steed, A., Goddard, R. et al. A wheat kinase and immune receptor form host-specificity barriers against the blast fungus. Nat. Plants 9, 385–392 (2023). https://doi.org/10.1038/s41477-023-01357-5
Minor, but important concern:
We were a little surprised by the following statement (lines 173-175):
“Shorter k-mers can be expected to provide more raw material for GWAS, as a smaller value of k results in greater k-mer counts and a lesser impact of sequencing errors.”
In our own experience with a >5 Gb plant genome, we have found that the number of 32-mers is smaller (less than half) than the number of 51-mers. Therefore, we suggest that the authors back up the statement with a reference to a study empirically underpinning their statement.
Typos:
Line 96: “a scale” should be “at scale”.
Line 178: There should be a full stop before “Typically”.
Line 234: “K-mer” should be “k-mer” (with lower case k).
Line 473: “guplication” should be “duplication”.
Author Response
##REVIEWER 1
#General comment: Karikari, Lemay and Belzile’s have written a welcome review on the application k-mer GWAS in plants, summarizing eloquently the advantages of k-mer GWAS and the challenges that lie ahead. The application of k-mers in genomics and population genetics is a rapidly emerging field – indeed one colleague of ours recently said “you can do almost anything with k-mers these days.” The review is well written, and the figures illustrate in an elegant and simple way some of the quintessential advantages of k-mer GWAS. We therefore expect the review will find traction amongst the scientific community studying plant genetics, genomics, and population biology. Notwithstanding, we do have some concerns, as listed below, which we would advise be addressed:
Response: Thank you for your valuable suggestion for us to improve the quality of our review. We have carefully revised the manuscript based suggestions by your good self and other three reviewers.
#Issue 1: Surprisingly, for such a new and emerging field, the review fails to make any mention of the pioneering work by Arora et al., 2019, Gaurav et al., 2022, and Arora et al., 2023. Arora et al., 2019 were the first to employ k-mer GWAS in plants, the first to demonstrate the use of k-mer GWAS without a reference genome, and the first to use the technology to clone plant genes. This feat was repeated in Gaurav et al., 2022, and in Arora et al., 2023. The three studies demonstrated the use of k-mer GWAS in two different species, the wild wheat progenitor Aegilops tauschii as well as hexaploid bread wheat, Triticum aestivum. To our knowledge, no other study has to date used k-mer GWAS to identify new genes for which the function was unequivocally confirmed using transgenics, mutants, and/or VIGS.
By not mentioning these studies in their review, Karikari, Lemay and Belzile falsely attribute ‘first’ achievements and critical advances to other studies and make false conclusions about the length of k-mers used in previous studies. We suggest the authors carefully go through their review, re-consider Table 1, and rewrite these critical statements considering this trio of studies.
Arora, S., Steuernagel, B., Gaurav, K. et al. Resistance gene cloning from a wild crop relative by sequence capture and association genetics. Nat Biotechnol 37, 139–143 (2019). https://doi.org/10.1038/s41587-018-0007-9
Gaurav, K., Arora, S., Silva, P. et al. Population genomic analysis of Aegilops tauschii identifies targets for bread wheat improvement. Nat Biotechnol 40, 422–431 (2022). https://doi.org/10.1038/s41587-021-01058-4
Arora, S., Steed, A., Goddard, R. et al. A wheat kinase and immune receptor form host-specificity barriers against the blast fungus. Nat. Plants 9, 385–392 (2023). https://doi.org/10.1038/s41477-023-01357-5
Response: Thank you for drawing our attention to some missing references in our earlier version of the manuscript. We have carefully read and added these references and their findings in the revised manuscript.
#Issue 2: Minor, but important concern:
We were a little surprised by the following statement (lines 173-175):
“Shorter k-mers can be expected to provide more raw material for GWAS, as a smaller value of k results in greater k-mer counts and a lesser impact of sequencing errors.”
In our own experience with a >5 Gb plant genome, we have found that the number of 32-mers is smaller (less than half) than the number of 51-mers. Therefore, we suggest that the authors back up the statement with a reference to a study empirically underpinning their statement.
Response: We realize that our initial phrasing was not clear enough. We actually meant the total number of k-mers counted from the dataset, which will necessarily be higher using smaller k, and not the number of unique k-mers. We have rephrased the corresponding sentences as follows and hope that it is clearer:
“Shorter k-mers reduce the probability of including sequencing errors and result in more k-mers being tallied for a given dataset, therefore providing higher counts for resolving copy number variants based on k-mer counts. On the other hand, longer k-mers would provide a larger number of unique k-mers and more power to resolve sequences stemming from repetitive regions.”
#Issue 3: Typos:
Line 96: “a scale” should be “at scale”.
Line 178: There should be a full stop before “Typically”.
Line 234: “K-mer” should be “k-mer” (with lower case k).
Line 473: “guplication” should be “duplication”.
Response: We have corrected the above typos in the revised manuscript.
Reviewer 2 Report
In this review, the authors described the advantage of k-mer-based GWAS, its concept, methods, applications, associated challenges and perspectives. My comments are:
1. Line 43. Linkage mapping is not equivalent to biparental mapping. Linkage mapping can also be applied to multi-parental populations. When multi-parental populations are used, multiple alleles can also be detected.
2. Section 1. In the second paragraph from bottom, there are some contents about the limitations of SNP-based GWAS, which are closer to the title of Section 2.
3. Title of Section 2 is limitation of SNP-based GWAS. But most contents are about quality of SVs. The first two sections are not well designed. I advise you to re-organize the first two sections. For example, the first section about the limitation, and the section about the SVs.
4. Line 234. You said that k-mer-based GWAS can be applied in species without reference genome. When there is no reference genome, how will step 3 be done (Figure 2c)? Can Lemay et al. [58] solve this problem? Make this point clear in main text.
5. Line 313. As you mentioned SV-GWAS several times, I advise you to introduce this method briefly in the main text.
6. List some traits from citation [52] in Table 1 instead of "several".
No comments.
Author Response
##REVIEWER 2
#General comment: In this review, the authors described the advantage of k-mer-based GWAS, its concept, methods, applications, associated challenges and perspectives. My comments are:
#Issue 1: Line 43. Linkage mapping is not equivalent to biparental mapping. Linkage mapping can also be applied to multi-parental populations. When multi-parental populations are used, multiple alleles can also be detected.
Response: We have removed the reference to biparental mapping and the claim of multiple alleles being an advantage of GWAS.
#Issue 2: Section 1. In the second paragraph from bottom, there are some contents about the limitations of SNP-based GWAS, which are closer to the title of Section 2.
#Issue 3: Title of Section 2 is limitation of SNP-based GWAS. But most contents are about quality of SVs. The first two sections are not well designed. I advise you to re-organize the first two sections. For example, the first section about the limitation, and the section about the SVs.
Response: We have addressed issues 2 and 3 by remodelling our first 2 sections. First, the title of these two sections was modified to better reflect their contents. The last paragraph of section 1 was then moved to section two and the section 2 was extensively modified to ensure a better flow. We believe that these sections should be more consistent and more easily readable now.
#Issue 4: Line 234. You said that k-mer-based GWAS can be applied in species without reference genome. When there is no reference genome, how will step 3 be done (Figure 2c)? Can Lemay et al. [58] solve this problem? Make this point clear in main text.
Response: We have added a whole paragraph at the end of section 4 describing how real-life workflows may differ from the one suggested, including a discussion of cases where a reference genome is not available.
#Issue 5: Line 313. As you mentioned SV-GWAS several times, I advise you to introduce this method briefly in the main text.
Response: We have addressed this point by adding the following sentence in section 2:
“The potential of a greater functional impact has motivated the use of SVs instead of SNPs for GWAS in crops such as cotton (Jin et al. 2023) soybean (Liu et al. 2019), cucumber (Li et al. 2022), and tomato (Dominguez et al. 2020), an approach which we refer to here as SV-based GWAS.”
#Issue 6: List some traits from citation [52] in Table 1 instead of "several".
Response: We have listed a few examples of traits used by Voichek and Weigel (2020) in the revised manuscript.

Reviewer 3 Report
The research paper titled “ k-mer-based GWAS in plants: advances, challenges, and per-2 spectives” submitted to Genes is well written and I recommend minor revision. The specific comments to review are given bellow;
Need to explain k-mers, in main text.
Fig. 1 needed add information to understand what is what? Not just draw lines
Line 163. Need to replace reference, and rewrite this sentences
Line 179; follow reference according to journal format
All figures hard to read, revise it
Line 245. Reference mistake
References are not uniform throughout the manuscript, must be consistent, and must be formatted according to the guidelines of the journal.
Please extend conclusions with future prospects and how these findings could be used for human benefits.
Author Response
##REVIEWER 3
#Comments and Suggestions for Authors
The research paper titled “ k-mer-based GWAS in plants: advances, challenges, and per-2 spectives” submitted to Genes is well written and I recommend minor revision. The specific comments to review are given bellow;
#Issue 1: Need to explain k-mers, in main text.
Response: The concept of k-mers was already defined in the first sentence of section 3. However, we have added an example for clarity:
“k-mers are subsequences of a fixed length k which can be obtained from sequencing data or genome assemblies by extracting all such subsequences found in the input dataset. For example, from the sequence ACCGTCG, the following k-mers of length four (4-mers) can be observed: ACCG, CCGT, CGTC, and GTCG.”
#Issue 2: Fig. 1 needed add information to understand what is what? Not just draw lines
Response: We have revised figure 1 and hope that it is clearer now.
#Issue 3: Line 163. Need to replace reference, and rewrite this sentences
Response: We have revised the sentences to bring clarity.
#Issue 4: Line 179; follow reference according to journal format
Response: We have formatted the references to according MDPI reference style for EndNote Reference software.
#Issue 5: All figures hard to read, revise it
Response: We have increased font size in figure 3. If any other changes are needed to improve readability, please let us know.
#Issue 6: Line 245. Reference mistake
Response: Lemay et al. (2023) was not wrongly cited. Could you please specify what the mistake is?
#Issue 7: References are not uniform throughout the manuscript, must be consistent, and must be formatted according to the guidelines of the journal.
Response: We used the reference style of MDPI available on the journal’s website, however any inconsistency could be rectified if our write-up is accepted for publication by the typesetter.
#Issue 8: Please extend conclusions with future prospects and how these findings could be used for human benefits.
Response: We now end the conclusion with this paragraph:
“Beyond methodological improvements in GWAS analysis, we believe that the use of k-mer-based association methods in the years to come will prove transformative for the discovery of new gene-trait associations. As discussed in this review, k-mer-based methods have already provided novel insights in crops such as soybean, maize, and wheat. Going forward, we believe that such approaches will greatly enhance our understanding of crop genetics and ultimately our ability to develop crop varieties that will tackle the challenges associated with population growth and climate change.”

Reviewer 4 Report
Genes Manuscript #: 247569 review
Authors: B. Karikari et al.
Title: k-mer-based GWAS in plants: advances, challenges, and perspectives
The authors have written a somewhat focused review article on the use of “k-mer” bioinformatic strategy to identify sequence variances among genomic sequence data that associate with phenotypes in plants. They argue the recent limited use of k-mer-based genome-wide associate studies (GWAS) in plants indicate the advantages and benefits it has to SNP-based only GWAS, since focusing on SNPs alone would miss other types of common genomic variances, such as: indels and structural variances (SV’s). They point out that although k-mer based approaches have advantages, there are still some key issues that need to be resolved/improved before this approach will be usable by a wider range of plant scientists to identify/map specific genomic variances and loci that associate with phenotypes.
In general, their arguments and points are both legitimate and needed, as genomic sequencing continues to be used more and more often and by a wider range of plant scientist, including those without primary expertise in bioinformatics. The writing is clear and well cited/referenced, which some minor issues I point out below. Overall, I feel that in general this review is useful and relevant.
However, in additional to some minor writing issues for clarity, I think there are three more significant aspects (varying levels) that the authors need to expand on and address more completely. First, there is need to include more information about the current and now common use of long-read genomic sequencing methods, such as Pacific Bioscience (PacBios) and Oxford Nanopore sequencing. The authors focus mostly on the analysis if genomic sequence data from short-read sequencing methods (Illumina), which is useful and important. But, PacBios and Oxford Nanopore sequencing methods are now common and routine when sequencing large genomes with repetitive DNA, such as most plants (and eukaryotes in general) when trying to map sequence variance. The authors include a very short sequence in the last paragraph (L369-377) that mentions the value and benefit of long-read sequencing methods and how this will allow the use of longer k-mers to be used, which would improve the accuracy and process. However, it is stated that the time when long-read sequencing data will be available is “…probably not too far.” (L377). In my opinion, for this review to be relevant now and into the future, in light of the fact that long-read sequencing is already here (and has been for several years), it should include more on long-read data for k-mer analysis. To support this is the fact, that several of the commonly cited references in this review include long-read sequence analysis (citations # 46 – 48). Related to this, the review should include a more details on how the k-mer methodology would change (or not) when using long-read sequence data, such as what would be the optimal k-mer lengths, how would this change the needed “depth of coverage” (if at all), etc...
The second somewhat significant aspect deals with Section 5, “Case studies…”: The authors point out many examples of specific findings in different plant genome studies in which using SNP-based analysis failed to find some key examples of variations (such as indels, inversions, repeats, etc…) that were found by k-mer methods, which makes sense for SNP analysis only focuses on SNPs and not the others, as the authors point out. However, I feel it would be good for the authors to either discuss specific case study examples were k-mer analysis might have missed what was found with SNP-based analysis. Several cited papers deal with this question directly, with citation #52 being one of these. The authors generally touch on this question (Lines 319-321) but without much detail. Expanding on this question seems needed for it is central to the question of which method is best. I would also ask the authors would there be value or specific situations when both k-mer AND SNP-based analysis should be done?
The third somewhat significant aspect deals with layout/format issues to Figures that would help with clarity and reader understand. I mention these specifically under the “Detailed/Specific Comments” section below.
Detailed/Specific Comments (mostly minor writing suggestions):
Line 17 (Abstract): I would either replace “which” for “that” OR add a common before, so it would read, “, which….”.
Line 19 (Abstract): Please fully write out/define “insertion/deletion (indel)”, since this is the first use of it in the review. Although indel is commonly known, as a review it would be best to fully define it, just like the authors correctly defined SNPs in Lines 14/15.
Line 33: Although it is fitting, the phrase “game changer” is a bit too jargony. But this is more a writing style issue.
Lines 58: Change “is” to “are”, since formally the word “data” is plural. However, this might also come down to a preference by Journal Editor.
Lines 68: Similar to above, I would either replace the word “which” for “that” OR add a common before, so it would read, “, which….”.
Lines 90 - 96: Here is the first section that sequence read length is emphasized. As was mentioned above, long-read sequencing methods are now commonplace. It would be good to mention this initially here, but then later expand on the use of long-read sequence data (as suggested above). In fact, this is the section that several of the cited references (#46 – 48) in fact emphasize long-read sequencing.
Line 126: Please fully write out/define “whole-genome sequencing (WGS)”, since this is the first use of it in the review, that I saw.
Line 127: Same as immediate above. Please fully write out/define genotype-by sequencing (GBS), since this is the first use of it in the review, that I saw.
Figure 1: I feel it would help the reader significantly if different colored lines were use to indicate the different specific k-mer in the figures to show which would be unique to different genomes/alleles/haplotypes in panels a, b, and c. That is, show different colored k -mer’s to the two different/specific genomes/alleles/haplotypes for panel a) to help readers see what is stated in the Figure Legend (L149-150). Then, use similar color differences for the different k -mer’s in panels b and c to highlight which k -mers would differ between the Indels (panel b) and inversions (panel c).
Figure 2: More explanation about what the “examples” show for the five different steps (across panels a – e) would help. That is, does these refer to bioinformatic programs/software, strategies, organisms or something different. More clarity on this is needed. And, please include a citation for each example, using #’s to be consistent with the reference/citations in the text. Finally, the Figure Legend could be expanded to help explain the examples, if that would be helpful.
Line 190: Same as above. Please fully write out/define multilevel model (MLM), since this is the first use of it in the review, that I saw.
Lines 270-272: I suggest adding some more explanation about what is in Figure 4 in the text. Not the alignment (panel a), but instead on panel b. Also, without adding too much space, it might be good to include an example for an indel or repeat and what real-time p-values for these would be.
Lines 288-289 and Table 1: The text states there were “several hundred” traits assessed for Arabidopsis, Maize and Tomatoes in citation #52 while in the Table (column three) indicates the same paper assessed “several” traits. From citation #52 directly, those authors wrote they “re-analyzed “2,000 traits” from Arabidopsis, Maize and Tomato. The text and Table 1 in this review need to be consistent between themselves as well is with the original cited paper.
Table 1: Add a white space/spacing between the “Trait” description for Maize and Soybean. As is, it is hard to tell where the traits list for Maize ends and trait list for Soybean begins.
Lines 348-358: It would seem that long-read sequencing methods (PacBios. Etc…) would help to significantly reduce the number of spurious associations in repetitive sequence regions. Again, the suggestion of an expanded discussion of now the now common long-read sequencing methods will much of this would be relevant here.
My comments above include some specific comments/suggestions regarding English language/writing.
Author Response
##REVIEWER 4
#General comment: The authors have written a somewhat focused review article on the use of “k-mer” bioinformatic strategy to identify sequence variances among genomic sequence data that associate with phenotypes in plants. They argue the recent limited use of k-mer-based genome-wide associate studies (GWAS) in plants indicate the advantages and benefits it has to SNP-based only GWAS, since focusing on SNPs alone would miss other types of common genomic variances, such as: indels and structural variances (SV’s). They point out that although k-mer based approaches have advantages, there are still some key issues that need to be resolved/improved before this approach will be usable by a wider range of plant scientists to identify/map specific genomic variances and loci that associate with phenotypes.
In general, their arguments and points are both legitimate and needed, as genomic sequencing continues to be used more and more often and by a wider range of plant scientist, including those without primary expertise in bioinformatics. The writing is clear and well cited/referenced, which some minor issues I point out below. Overall, I feel that in general this review is useful and relevant.
However, in additional to some minor writing issues for clarity, I think there are three more significant aspects (varying levels) that the authors need to expand on and address more completely. First, there is need to include more information about the current and now common use of long-read genomic sequencing methods, such as Pacific Bioscience (PacBios) and Oxford Nanopore sequencing. The authors focus mostly on the analysis if genomic sequence data from short-read sequencing methods (Illumina), which is useful and important. But, PacBios and Oxford Nanopore sequencing methods are now common and routine when sequencing large genomes with repetitive DNA, such as most plants (and eukaryotes in general) when trying to map sequence variance. The authors include a very short sequence in the last paragraph (L369-377) that mentions the value and benefit of long-read sequencing methods and how this will allow the use of longer k-mers to be used, which would improve the accuracy and process. However, it is stated that the time when long-read sequencing data will be available is “…probably not too far.” (L377). In my opinion, for this review to be relevant now and into the future, in light of the fact that long-read sequencing is already here (and has been for several years), it should include more on long-read data for k-mer analysis. To support this is the fact, that several of the commonly cited references in this review include long-read sequence analysis (citations # 46 – 48). Related to this, the review should include a more details on how the k-mer methodology would change (or not) when using long-read sequence data, such as what would be the optimal k-mer lengths, how would this change the needed “depth of coverage” (if at all), etc...
Response: Given the fact that k-mer-based GWAS has not yet been used with long-read data and that long-read data is probably not yet ready for such use, we chose not to discuss the use of long reads in sections 3-5, which aim to provide guidelines or report the results of previous studies. However, based on this comment, we expanded the paragraph of section 6 in which we already discussed long reads. This paragraph now reads:
“Finally, we anticipate that future developments in sequencing technology should enhance the use of k-mers in GWAS. So far, sufficient sequencing depth and sequence quality for k-mer-based analyses has only been provided by the Illumina short-read sequencing technology. As a result, k-mer length for use in GWAS is limited to a few dozen nucleotides, whereas k-mer lengths in excess of a hundred nucleotides would provide much greater resolution of variants located in repetitive regions. The relatively high error rates of current long-read technologies (PacBio and Oxford Nanopore technologies, Amarasinghe et al. 2020) prohibit the extraction of long k-mers from their sequenc-es, as the probability of including errors in such k-mers would be extremely high. One exciting development is with increases in the availability and quality of long-read sequencing technologies (such as PacBio HiFi data [102]), which can deliver highly accurate long reads; however, this method is still too costly to apply at the population scale required for GWAS. In the meantime, the use of long-read correction methods (Zhang et al. 2020) may be an interesting avenue for the use of long-read sequencing data in k-mer. Overall, given the pace of recent developments in long-read technologies, the moment when long-read sequencing will fuel k-mer-based GWAS efforts is probably not too far.”
The second somewhat significant aspect deals with Section 5, “Case studies…”: The authors point out many examples of specific findings in different plant genome studies in which using SNP-based analysis failed to find some key examples of variations (such as indels, inversions, repeats, etc…) that were found by k-mer methods, which makes sense for SNP analysis only focuses on SNPs and not the others, as the authors point out. However, I feel it would be good for the authors to either discuss specific case study examples were k-mer analysis might have missed what was found with SNP-based analysis. Several cited papers deal with this question directly, with citation #52 being one of these. The authors generally touch on this question (Lines 319-321) but without much detail. Expanding on this question seems needed for it is central to the question of which method is best. I would also ask the authors would there be value or specific situations when both k-mer AND SNP-based analysis should be done?
Response: When discussing the results of Voichek and Weigel (2020), we added the following sentence:
“Nevertheless, some significant associations were only detected through SNP-based analysis, suggesting that SNP-based analysis may be complementary to k-mers.”
We also added the following sentence when discussing the results of Lemay et al. (2023):
“This study therefore advocated for the complementarity of SNP- and k-mer-based approaches in identifying candidate genes.”
The third somewhat significant aspect deals with layout/format issues to Figures that would help with clarity and reader understand. I mention these specifically under the “Detailed/Specific Comments” section below.
Response: Thank you for your valuable suggestion for us to improve the quality of our review. We have carefully revised the manuscript based on your suggestions and those of three other reviewers.
Detailed/Specific Comments (mostly minor writing suggestions):
Line 17 (Abstract): I would either replace “which” for “that” OR add a common before, so it would read, “, which….”.
Response: We have modified as you suggested.
Line 19 (Abstract): Please fully write out/define “insertion/deletion (indel)”, since this is the first use of it in the review. Although indel is commonly known, as a review it would be best to fully define it, just like the authors correctly defined SNPs in Lines 14/15.
Response: We have replaced ‘indel’ with ‘insertions/deletions’ in the revise manuscript.
Line 33: Although it is fitting, the phrase “game changer” is a bit too jargony. But this is more a writing style issue.
Response: We have modified to ‘Genome-wide association studies (GWAS) have routinely been used in plant science……….’
Lines 58: Change “is” to “are”, since formally the word “data” is plural. However, this might also come down to a preference by Journal Editor.
Response: We have replaced ‘is’ with ‘are’ in the revised manuscript.
Lines 68: Similar to above, I would either replace the word “which” for “that” OR add a common before, so it would read, “, which….”.
Response: This section was modified, so this comment is no longer relevant.
Lines 90 - 96: Here is the first section that sequence read length is emphasized. As was mentioned above, long-read sequencing methods are now commonplace. It would be good to mention this initially here, but then later expand on the use of long-read sequence data (as suggested above). In fact, this is the section that several of the cited references (#46 – 48) in fact emphasize long-read sequencing.
Response: See our response to the issue mentioned above.
Line 126: Please fully write out/define “whole-genome sequencing (WGS)”, since this is the first use of it in the review, that I saw.
Response: ‘WGS’ was fully defined on Line 58 in the earlier version of the manuscript.
Line 127: Same as immediate above. Please fully write out/define genotype-by sequencing (GBS), since this is the first use of it in the review, that I saw.
Response: ‘GBS’ was written fully on Line 60 in the earlier version of the manuscript.
Figure 1: I feel it would help the reader significantly if different colored lines were use to indicate the different specific k-mer in the figures to show which would be unique to different genomes/alleles/haplotypes in panels a, b, and c. That is, show different colored k -mer’s to the two different/specific genomes/alleles/haplotypes for panel a) to help readers see what is stated in the Figure Legend (L149-150). Then, use similar color differences for the different k -mer’s in panels b and c to highlight which k -mers would differ between the Indels (panel b) and inversions (panel c).
Response: We have revised figure 1 and hope that it is clearer now.
Figure 2: More explanation about what the “examples” show for the five different steps (across panels a – e) would help. That is, does these refer to bioinformatic programs/software, strategies, organisms or something different. More clarity on this is needed. And, please include a citation for each example, using #’s to be consistent with the reference/citations in the text. Finally, the Figure Legend could be expanded to help explain the examples, if that would be helpful.
Response: We have added reference numbers in the figure and expanded the figure caption to provide more explanations.
Line 190: Same as above. Please fully write out/define multilevel model (MLM), since this is the first use of it in the review, that I saw.
Response: We have replaced ‘MLM’ with the right word ‘mixed linear model’ in the revised manuscript.
Lines 270-272: I suggest adding some more explanation about what is in Figure 4 in the text. Not the alignment (panel a), but instead on panel b. Also, without adding too much space, it might be good to include an example for an indel or repeat and what real-time p-values for these would be.
Response: We added an example for an indel as requested. We would be happy to add more explanations regarding panels b and d, but would need more details on what is unclear.
Lines 288-289 and Table 1: The text states there were “several hundred” traits assessed for Arabidopsis, Maize and Tomatoes in citation #52 while in the Table (column three) indicates the same paper assessed “several” traits. From citation #52 directly, those authors wrote they “re-analyzed “2,000 traits” from Arabidopsis, Maize and Tomato. The text and Table 1 in this review need to be consistent between themselves as well is with the original cited paper.
Response: We have added some of the traits from the paper of Voichek and Weigel (2020) in the revised manuscript.
Table 1: Add a white space/spacing between the “Trait” description for Maize and Soybean. As is, it is hard to tell where the traits list for Maize ends and trait list for Soybean begins.
Response: We have modified as you suggested.
Lines 348-358: It would seem that long-read sequencing methods (PacBios. Etc…) would help to significantly reduce the number of spurious associations in repetitive sequence regions. Again, the suggestion of an expanded discussion of now the now common long-read sequencing methods will much of this would be relevant here.
Response: See the response above.

Round 2
Reviewer 1 Report
The authors have addressed the concerns we had with the first submission. We feel the manuscript has been substantially improved and have no further concerns. In conclusion, we are now in full support of publication.
Reviewer 2 Report
I have no additional comments.